# Investigating Potential Cancer Therapeutics: Insight into Histone Deacetylases (HDACs) Inhibitions

**DOI:** 10.3390/ph17040444

**Published:** 2024-03-29

**Authors:** Basharat Ahmad, Aamir Saeed, Ahmed Al-Amery, Ismail Celik, Iraj Ahmed, Muhammad Yaseen, Imran Ahmad Khan, Dhurgham Al-Fahad, Mashooq Ahmad Bhat

**Affiliations:** 1School of Life Science and Technology, Center for Informational Biology, University of Electronics Science and Technology of China, Chengdu 610056, China; 2Department of Bioinformatics, Hazara University Mansehra, Mansehra 21120, Pakistan; 3Department of Physiology and Medical Physics, College of Medicine, University of Thi-Qar, Nasiriyah 64001, Iraq; 4Department of Pharmaceutical Chemistry, Faculty of Pharmacy, Erciyes University, 38280 Kayseri, Turkey; ismailcelik@erciyes.edu.tr; 5Atta-Ur-Rehman School of Applied Biosciences (ASAB), National University of Science and Technology (NUST), Islamabad 44000, Pakistan; iraj.malik921113@gmail.com; 6Institute of Chemical Sciences, University of Swat, Charbagh 19130, Pakistan; muhammadyaseen.my907@gmail.com; 7Department of Chemistry, Government College University Faisalabad, Faisalabad 38000, Pakistan; imrankhan707470@gmail.com; 8Department of Pharmaceutical Sciences, College of Pharmacy, University of Thi-Qar, Nasiriyah 64001, Iraq; dhurgham.alfahad@sci.utq.edu.iq; 9Department of Pharmaceutical Chemistry, College of Pharmacy, King Saud University, Riyadh 11421, Saudi Arabia

**Keywords:** histone deacetylases, neuroblastoma, molecular docking, molecular dynamic simulation

## Abstract

Histone deacetylases (HDACs) are enzymes that remove acetyl groups from ɛ-amino of histone, and their involvement in the development and progression of cancer disorders makes them an interesting therapeutic target. This study seeks to discover new inhibitors that selectively inhibit HDAC enzymes which are linked to deadly disorders like T-cell lymphoma, childhood neuroblastoma, and colon cancer. MOE was used to dock libraries of ZINC database molecules within the catalytic active pocket of target HDACs. The top three hits were submitted to MD simulations ranked on binding affinities and well-occupied interaction mechanisms determined from molecular docking studies. Inside the catalytic active site of HDACs, the two stable inhibitors LIG1 and LIG2 affect the protein flexibility, as evidenced by RMSD, RMSF, Rg, and PCA. MD simulations of HDACs complexes revealed an alteration from extended to bent motional changes within loop regions. The structural deviation following superimposition shows flexibility via a visual inspection of movable loops at different timeframes. According to PCA, the activity of HDACs inhibitors induces structural dynamics that might potentially be utilized to define the nature of protein inhibition. The findings suggest that this study offers solid proof to investigate LIG1 and LIG2 as potential HDAC inhibitors.

## 1. Introduction

Epigenetic alterations are crucial for the development of malignancies and the growth of tumors. These changes include non-coding RNAs (ncRNAs), the methylation of DNA, and modifications of histones (acetylation, phosphorylation, and methylation) [1]. Despite the different modifications that have been investigated, histone acetylation has been widely studied. Histone acetyltransferases (HATs) and histone deacetylases (HDACs) constitute two groups of enzymes that keep histones acetylated [2]. Both enzymes have an impact in altering the chromatin’s structure and accessibility to the transcription factors [3]. Therefore, histone deacetylases (EC 3.5.1.98, HDAC) and acetyltransferases (EC 2.3.1.48, HAT) are enzymes that add and remove the acetyl groups of lysine residues, respectively [4]. Surprisingly, abnormal activity of the HDAC enzymes has been discovered in a wide range of cancer types. Therefore, HDAC has been evaluated as a possible therapeutic target for cancer treatment [5]. The histone deacetylases (HDACs) are zinc metalloenzymes that catalyze the removal of acetyl group moieties from the ε-amino groups of lysine residues on the amino-terminal tails of core histones to regulate gene transcription and chromatin remodeling [6]. HDAC-deacetylated histones with positively charged N-terminal tails bind with DNA’s negatively charged phosphate groups [7]. HDACs are important regulators of gene expression, cell development, proliferation, and their over-expression involved in the development of human malignancies [8]. Hence, chromatin has been compacted into a condensed form (heterochromatin) related to lower gene transcriptions, and HAT activity can reverse this structural state, resulting in the transcriptionally active DNA (euchromatin) being relaxed. Therefore, HAT/HDAC activity is balanced to define histone acetylation levels, which play a significant function in gene transcription regulations via epigenetics changes. A variety of chronic diseases such as inflammation, neurological, diabetes, and cancer disorders have been correlated with changes in this highly coordinated biological system [9,10]. In humans, a total of eighteen HDAC isoforms have currently been discovered so far and categorized into four groups based on their similarity to yeast HDACs. Class I (HDAC 1–3, and 8), class II (HDAC 4–7, 9 and 10), class III (SIRT 1–7), and class IV (HDAC11) are Zn^2+^-dependent metallohydrolases belonging to class I, II, and IV, whereas NAD+-dependent Sir2-like deacetylases belong to class III [11]. Of note, HDAC1, 2, and 3 are members of the class I isoform family, which is evolutionarily conserved. It is highly expressed in the majority of solid and hematologic malignancies and is significantly associated with a poor prognosis, although it is absent from calm endothelium cells and healthy organs [12]. Surprisingly, HDAC1, 2, and 3 have garnered significant interest due to their diverse biological roles that are essential in both normal and malignant cells. As distinct members belonging to the class I HDAC enzyme group, HDAC1, HDAC2, and HDAC3 possess a common characteristic in their catalytic domain, consisting of one histidine (His) and two aspartate (Asp) residues. These residues encompass a Zn^2+^ ion co-factor, making it easier for Zn^2+^ to hydrolyze ε-N-acetylated lysine residues. This process leads to histone deacetylation via a charge relay mechanism [13]. HDAC1, HDAC2, and HDAC3 exhibit not just nuclear localization but also wide expression, implying a potentially key involvement in carcinogenesis [14,15,16]. Inhibitors of HDAC were identified as an emerging category of chemotherapeutic agents. Increased histone acetylation induced by histone deacetylase inhibitors (HDACIs) promotes cellular activities such as cell cycle progression, division, and death [17]. A growing amount of information about the associations between HDAC inhibition and its potential applications in cancer treatment has surfaced in recent years. Numerous studies have been carried out to investigate a wide variety of HDAC inhibitors, which include both natural and synthetic compounds. Currently, several histone deacetylase inhibitors (HDACIs) such as vorinostat and romidepsin were recently approved for the therapy of cutaneous T cell lymphoma [18]. Meanwhile, various additional HDACIs continue to be studied in clinical trials. Histone deacetylase inhibitors (HDACIs) possess several anticancer mechanisms that include inducing apoptosis, arresting the cell cycle, inhibiting angiogenesis, producing reactive oxygen species (ROS), and initiating DNA damage [19]. Therefore, class-selective HDACIs and isoforms are in high demand to reduce adverse reactions from off-target activity. According to X-ray crystallographic research, an HDAC-like protein (HDLP) associated with SAHA was found in the hydrophobic pocket that included a tube-like internal channel of 11 Å and a core region containing Zn^2+^ found at the bottom of the internal channel [20]. Metal binding (ZBG), linker, and surface recognition (CAP) are three different domains found in HDAC inhibitors [21]. The metal binding domain seems to be a prerequisite for the inhibitory action of HDAC because it chelates with Zn^2+^ in the active site of HDACs, while the linker domain requires hydrophobicity. Similarly, surface recognition domains are required for enzymes to recognize and attach within the rim of the binding site [22].

In recent years, there has emerged a rising tendency to emphasize the utilization of computer-aided drug design (CADD) within the process of drug development and discovery. Both ligand- and structure-based drug designs were encompassed by CADD approaches [23,24]. Their combinatorial application alongside high-throughput screening has been shown to accelerate drug discovery and optimization processes significantly [25,26]. Molecular docking and molecular dynamics have been proven to play a crucial role in the virtual screening of novel compounds for drug development [27,28]. Recently, rational HDAC inhibitors using docking coupled with molecular dynamics protocol have been more beneficial in improving docking results and offering insights into inhibitory processes [29].

Considering the inhibitory activities of compounds, we focused on determining compounds as promising HDAC1, 2, and 3 selective inhibitors in terms of finding more promising inhibitors for anti-cancer therapy through high-throughput screening via molecular docking coupled with molecular dynamic simulation protocols. In this study, a pool of library compounds from multiple databases was subjected to virtual screening against HDAC1, HDAC2, and HDAC3 to efficient HDAC inhibitors that may be explored further as anticancer drugs. We presented the computational study and screened the top binders’ inhibitors which interact in the catalytic site of the Zn^2+^-binding group. Top binding modes and selectively binding pockets of HDAC enzymes were appropriately assessed by scoring functions, and inhibitors exhibiting reasonably high s-scores were submitted to molecular dynamic simulation to examine the interaction patterns of these inhibitors at the atomic level.

## 2. Results and Discussion

### 2.1. Database Screening and Virtual Screening

To gain insight into the atomistic scale-binding mechanism between the HDAC enzymes and inhibitors, the molecular docking approach was utilized. The docking compounds were ordered based on a stringent filter that includes four factors: the highest occupation of the catalytic active pocket with minimum Gibbs free energy, hydrogen bond propensities, and additional potentially non-covalent interactions cumulatively estimated and represented with an S-core function [30]. The top-ranking docking poses were determined from among 5000 docked compounds. The ranking criterion comprised thresholds that needed a ligand to exhibit the necessary S-score values (the lower the s-score, the better the binding affinities and interactions) in association with HDACs targets incorporating all the hotspot residues of the catalytic binding sites. As observed in the crystal structure of HDAC2 (PDB: 7LTG) bound to apicidin accommodating a ketone Zn-binding element, the ketone region of bound apicidin with the catalytic Zn^2+^ ion of HDAC2 in the form of hydrated gem-diol(ate), establishing two chelation with the Zn^2+^ ion. The coordinating distances existed between the Zn^2+^ ion and the pair of oxygen atoms of the gem-diol(ate) with bond lengths of 1.9 Å and 2.4 Å, exhibiting asymmetries. Compared with the co-crystalized apicidin and HDAC2 docking complexes, the gem-diol(ate) of the oxygen atoms across ketone interacted with Zn^2+^ at distances measured at 1.9 Å and 2.6 Å, as shown in Figure 1B. Moreover, the compound library of the ZINC database was docked inside the specified binding pocket of all three HDACs enzymes, whereas the co-crystalized ligand apicidin was treated as a control for HDAC2 to determine the exact conformations. All the docking complexes were screened based on the highest binding score and notable interaction mechanisms with the active site residues and chelating with the Zn^2+^ ion. Based on the selection criteria, Table 1 displays the eight best promising inhibitors in terms of binding affinity and interaction mechanism with HDAC enzymes (Figure 2). Three inhibitors including LIG1 (ZINC98207834), LIG2 (ZINC77024375), and LIG3 (ZINC67801495) were found to exhibit the highest binding score and formed a metal coordination with Zn^2+^ within the catalytic site of the HDAC enzymes. The binding s-score (−19.1103, −27.0314, −27.4039 Kcal/mol) of LIG1 usually ranked at the top of the list when it came to binding energy, as evaluated by the s-score, followed by LIG2 (−21.8915, −23.2849, −27.9976 Kcal/mol) and LIG3 (−17.5725, −20.1273, −16.1137 Kcal/mol) having significantly lower binding energies in comparison to LIG1 (Table 1). The binding mode between inhibitors and HDAC enzymes, hydrogen bonds, and hydrophobic interactions play a critical role in stabilizing the protein–ligand complexes and increasing the binding energies [31,32].

### 2.2. Molecular Docking Interactions

To understand the binding mode of the top three hit protein–ligand complexes, we elucidated the binding mechanisms, like hydrogen bonds and hydrophobic interactions. As depicted in Figure 3, all these drug-like compounds docked effectively into the catalytic active site. LIG1 exhibited a noteworthy binding profile with the target HDAC1, wherein the oxygen atom established hydrogen bonding interactions with H178 at 2.5 Å. In parallel, LIG1 established coordination with Zn^2+^ at a proximity of 1.5 Å. In addition, a π-interaction was also observed as being bound by the residue F205 near the close moiety of the iodobenzene region. A similar binding interaction was shown in the HDAC2 binding pocket: the oxygen atom of the inhibitor was engaged via hydrogen bonding with H179 (2.6 Å) and chelated with Zn^2+^(1.0 Å). However, this interaction pattern displayed two more hydrogen bonds bound with the imidazole of sidechain H141 (-NH_3_) and H142 (-COOH) with distances of 2.4 Å and 2.2 Å, respectively. Additionally, the binding mechanism of LIG1 bound with HDAC3 was observed with three hydrogen bonds of H135, H172, and Y298 with oxygen, -COO, and -NH groups at distances of 2.6 Å, 3.0 Å, and 1.7 Å. Zn^2+^ was chelated by both the -COO group and oxygen atom of the inhibitor at distances of 2.7 Å and 1.3 Å. Furthermore, the interaction profile of compound LIG2 with HDAC1 demonstrated the chelation of Zn^2+^ at a distance of 1.3 Å, near to the vicinity of oxygen. Moreover, hydrogen bonding and π-interaction were observed with the imidazole region of H178 (2.7 Å), as well as F150 and F205. Similarly, the interaction of LIG2 bound with HDAC2, exhibiting a hydrogen bond facilitated by the residues H179 with a bond length of 1.5 Å, and the π-interaction was observed via hydrophobic residues F151 and F206. On the other hand, oxygen and nitrogen could form a chelate with Zn^2+^ with distances measured at 1.1 Å and 2.1 Å. Likewise, the imidazole moiety of H172 was involved in the hydrogen bond interaction with an oxygen atom at 2.0 Å, within the LIG3-HDAC3 complex. Simultaneously, the nitrogen atom of LIG3 formed a chelate with Zn^2+^ at distances measured at 1.0 Å and 2.0 Å, respectively. Hydrophobic residues F144 and F200 were also observed in the same pattern of the interaction, forming π-interactions. Moreover, the LIG3-HDAC1 complex exhibited two hydrogen bonds, and π-interactions within the close vicinity of the imidazole region of H141 (1.9 Å) bound with ethylene oxide, and the phenolic region of Y303 (1.8 Å) bound with oxygen of carbonyl. Residues F150 and H178 displayed π-interactions with the bromobenzene moiety of the inhibitor. Additionally, amino acid residues of HDAC2 bound with LIG3 were observed with a large number of hydrogen propensities. The ethylene oxide moiety of the inhibitor bound with two hydrogen bonds of imidazole H141 and H142 at bond lengths 2.5 Å and 2.2 Å, respectively. Moreover, the imidazole of H179 and phenolic moiety of Y304 established hydrogen bonds with oxygen and carbonyl groups at distances of 2.0 Å and 2.9 Å, respectively. The bromobenzene moiety of the inhibitor formed two π-interactions with Y151 and H179, respectively. Significantly, the interaction between LIG3 with target HDAC3 demonstrated the two π-interactions with Y144 and H172 at the region of the bromobenzene of the inhibitor. Meanwhile, upon introducing hydrogen bonds in this interaction pattern, the imidazole of H135 was observed within the close vicinity of oxygen at a distance of 2.7 Å. The interactions of the Zn^2+^ ion of all the HDACs were targeted in a complex with LIG3, oxygen formed a chelate with Zn^2+^ in the active pocket of HDAC1 at a distance of 1.3 Å, while carbonyl formed a coordination with Zn^2+^ in the active pocket of HDAC2 at a distance of 2.5 Å. Furthermore, both the oxygen and carbonyl group coordinated with the Zn^2+^ ion of the HDAC3 active pocket with closely related distances of 1.4 Å and 2.5 Å, respectively. All the interaction patterns of each complex are mentioned and represented in Table 2 and Figure 3. Several research studies have demonstrated that the addition of nitrogen-containing heterocyclic groups, such as thiazole, pyrazole, pyrimidine, thiadiazole, and triazole, to the surface recognition domain of HDACIs, can significantly enhance their selectivity and inhibitory potency [33]. It has been observed that the N, O, and S atoms, along with the electron-rich nitrogen-containing heterocycle, are capable of forming π–π and hydrogen bonding interactions with the amino acid residues located at the periphery of HDAC catalytic sites [34].

### 2.3. Molecular Dynamics Simulations

Molecular dynamic simulations constitute an extremely useful technique for fully comprehending the stability of protein–ligand-bound complexes [35,36]. In the present study, we performed molecular dynamics simulation for a 100 ns timeframe, after molecular docking, to validate and comprehensively investigate the selective top bonded compounds identified as possible inhibitors associated with HDACs targets. This study aimed to assess the complex’s dynamic motions, examine trajectory patterns, investigate structural characteristics, evaluate binding mechanisms, and identify potential conformational changes that may occur. The trajectory patterns were examined after 100 ns of the MD simulations system for each complex, monitoring the root mean square deviations (RMSD), root mean square fluctuations (RMSF), radius of gyration (Rg), and hydrogen bonds during the simulation system.

### 2.4. Root Mean Square Deviation Assessment

The RMSD offers a valuable measure for assessing protein and ligand structural stability due to the fact it quantifies the amount of atomic position deviation from the original position [37]. The RMSD of Cα, C, and N atoms of docked complexes were observed to evaluate the stability of trajectories during 100 ns of the simulation system. Minor oscillations and stable backbone atoms of RMSD indicate system stability [38]. As shown in Figure 4A, after an initial phase of oscillations, each complex reached a state of equilibrium and remained stable during 100 ns of simulation, demonstrating that the system folded into a more stable form than the initial structure. Compound LIG1 in a complex with HDAC1 achieved an equilibrium state after 50 ns of simulation, with RMSD values between 0.2 nm and 0.3 nm, respectively. In contrast, the LIG2-HDAC1 complex remained stable during the entire 100 ns simulation, maintaining a consistent RMSD value of 0.1 nm. However, there was a notable fluctuation between 60 and 65 ns, likely due to conformational changes resulting from the ligand’s precise fit into the catalytic site (Figure 4A).

After an initial leap owing to protein relaxation, the LIG1-HDAC2 complex attained stability after 55 ns with an RMSD around a mean value of 0.3 nm till the simulation was completed. Furthermore, the LIG2-HDAC2 complex stayed in a stable state from 1 ns to 50 ns with an RMSD value of 0.1 nm, and after that, it slightly increased its RMSD up to 0.15 nm till the simulation ended. Simultaneously, compound LIG1, in a complex with HDAC3, equilibrates at the initial phase from an RMSD value of 0.1 nm up to 25 ns and then increases its RMSD value up to 0.2 nm to 63 ns and then reaches back to its stabilized state till the simulation ends. In the case of LIG2-HDAC3, the complex showed significant flexibility from 28 to 38 ns and stabilized at an average RMSD value of 0.1 nm throughout the 100 ns of the timeframe (Figure 4B). As a result, in comparison with LIG1, compound LIG2 showed a better stable state with slight fluctuations and bound with all HDAC enzymes. These findings indicate that the dynamic stabilities of the complexes were reliable, and the trajectories might be essential in capturing snapshots for further investigation. Furthermore, the system’s stability added to the credibility of the docking results. Moreover, the RMSD graphs of the top binders bound with HDAC enzyme complexes were plotted with distinct colors, as shown in Figure 4A,B.

### 2.5. Protein Compactness Analysis

The radius of gyration (Rg) provides a parameter describing the protein’s total mean dimensions, with variations in this parameter suggesting the compression or expression of the complexes across the simulation system. The results indicate that the structural compactness of the enzyme has a considerable impact on the quality of the ligand interaction within the catalytic region. The variations effects within the Rg of HDAC enzymes bound to selected inhibitors LIG1 and LIG2 are depicted in Figure 4C. As shown in this figure, the LIG1 bound with all HDAC enzymes increased its Rg value in the initial phase from 1.95 nm and 1.97 nm to 2.05 nm and 2.06 nm and 1.96 nm up to 42,000 ps (42 ns), and after that, it became stable till the simulation ended. In comparison to LIG1 complexes, the HDACs’ Rg values (with a mean of 1.93, 1.94, and 1.95 nm) with LIG2-complexed systems equilibrated up to 100,000 ps (100 ns) significantly, which indicated protein structural compression after interacting with ligands. When it comes to Rg value fluctuations induced from alternating the release of protein compression, more oscillations might be observed in the Rg graph plots of a protein complex with LIG1 in comparison with LIG2. The Rg analysis revealed that the bound HDAC complexes have steady behavior and strong compactness, indicating that the identified possible inhibitors were firmly bound upon chelating Zn^2+^ and hydrogen bonds in the catalytic site of HDAC targets.

### 2.6. Hydrogen Bonds Analysis

Hydrogen bonds are crucial variables in protein–ligand-binding interactions. Hydrogen bonding, electrostatic, and hydrophobic interactions are different kinds of interactions in the protein–ligand complexes, but out of all of them, hydrogen bonds are particularly specialized interactions that contribute to the stabilizing of protein–ligand complexes. Along with protein structural stability, hydrogen bonds play a vital role in the ligand binding of LIG1 and LIG2 at the catalytic site of HDACs. Furthermore, the binding patterns were also evaluated by observing the fluctuations of the hydrogen bonding in all protein–ligand complexes. The maximum propensities of hydrogen bonding per time over all complexes during 100 ns of the molecular dynamic simulation are shown in Figure 4D. The outcomes suggest the development of a maximum of four, five, and six hydrogen bonds of LIG1 in association with HDAC1, 2, and 3 enzymes during the molecular dynamic simulation of the 100 ns timeframe. In the complex with HDAC enzymes, LIG2 established a maximum of two hydrogen bonds. These bonding factors revealed that LIG1 and LIG2 were successfully and firmly associated with all HDACs. Overall, the selected HDAC inhibitors demonstrated important hydrogen bonding interactions with active site residues of HDAC1, HDAC2, and HDAC3 enzymes, highlighting their potential as alternative therapeutic candidates.

### 2.7. Solvent Accessible Surface Area (SASA)

The solvent accessible surface area (SASA) parameter was performed to examine the magnitude as well as protein conformational changes upon ligand binding with the receptor. Additionally, it also refers to the surface area of a protein that interacts with its solvent molecules. The average SASA values for LIG1 and LIG2 bound with all the HDAC targets were determined throughout the 100 ns of the MD simulation. The current study found that LIG1 bound to HDAC2 and HDAC3 contributed to an average SASA value of 175 nm^2^, indicating similar a magnitude of protein structural changes following interactions with their respective compounds. Moreover, the analysis of SASA values for LIG1 bound with HDAC3 revealed lower degrees compared to HDAC1 and HDAC2. However, sudden oscillations occurred after 25 ns, leading to an increase in SASA value with an average of 165 nm^2^ until the simulation ended. Likewise, the binding of LIG2 with HDAC1 and HDAC2 showed a consistent pattern of SASA values reaching a steady range between 155 nm^2^ and 160 nm^2^, respectively. Furthermore, LIG2 bound with HDAC3 had the lowest SASA values, stabilizing between 150 nm^2^ and 155 nm^2^, respectively, as shown in Appendix A. Following ligand binding, no significant changes were observed in SASA values for HDAC1 and HDAC2, whereas HDAC3 exhibited lower SASA values and fluctuated around a constant value. As a result, it was concluded that 100 ns of simulation time was sufficient to bring the systems under investigation to equilibrium.

### 2.8. Protein Residues Fluctuation Analysis

The root mean square fluctuation (RMSF) of every amino acid residue is probably the most significant factor determining the ligand stability in the active site of target proteins [39]. The flexibility within the docked complexes was investigated to track the fluctuations of the residues. In this study, the flexibility of residues was examined using the RMSF of back-bone atoms for all complexes. As shown in Figure 5, compounds LIG1 and LIG2 had similar RMSF graphs in complexes with their respective HDAC enzymes. Most notably, the RMSF assessment of the associated LIG1 and LIG2 with HDAC enzymes demonstrates the flexibility of the key areas and reveals that the rotations of mobile loops significantly affected the ligand binding to HDACs. A notable difference appeared between the RMSFs of LIG1-bound HDAC enzymes in comparison to LIG2, as depicted in Figure 5. In contrast to LIG1-HDACs complexes, LIG2 possessed lower RMSF values. According to RMSF, the HDAC1, 2, and 3 electrostatic surfaces could be divided into different regions (larger and smaller regions) that showed fewer fluctuations throughout MD simulations. The RMSF value increased at various regions of amino acid residues, 17–33, 79–86, 100–110, 141–153, 176,182, 196–224, 261–176, and 300–305, in a complex with LIG1-HDAC1 (Figure 5). Larger loop regions C, F, and G showed higher fluctuations in the LIG1-HDAC1 complex with an RMSF value less than 0.4 nm. In contrast to the LIG1-HDAC2 complex, fewer fluctuations in regions (21–33, 196–224, and 270–274) were observed during the simulation. The specific region exhibited substantially heightened fluctuations, as shown by the RMSF spanning between 0.5 and 0.6 nm (Figure 5).

Moreover, the LIG1-HDAC3 complex showed higher fluctuations at B (196–226) and C (261–271) regions with the RMSF values 0.3 and 0.35 nm. Both regions were adjacent to the catalytic site (Figure 5). In the case of LIG2-HDACs, complexes fluctuation increased in the initial and final phase of the simulation at N and C terminal regions as depicted in Figure 6. These complexes show more stability in their loop regions compared with LIG1 in a complex with HDAC enzymes. Similarly, the LIG2-bound complexes comprised a few regions (97–101 and 371–376 region in LIG2-HDAC1 complex, 22–27,91–101,196–210,256–266 and 341–347 regions in LIG1-HDAC2 complex, 22–31, 95–100, 200–206, and 260–276 regions in LIG1-HDAC3 complex) where they showed some fluctuations during the simulation system (Figure 5). In the LIG2-HDAC1 complex, region A (97–101) was the only region that had higher fluctuations, with an RMSF value of 0.2 nm, followed by LIG2 in a complex with HDAC2, which showed a similar trend in region B (91–101), with an RMSF < 0.2 nm. In comparison with these two complexes, the LIG2-HDAC3 complex showed more fluctuations with a higher RMSF value of 0.3 nm at regions B (95–100) and E (341–347) (Figure 6). The binding mode was favored and more stable because LIG2 continued to interact with Zn^2+^ in the catalytic site of HDACs enzymes. Residues (H178, F150, and F205 of HDAC1, H179, F151, and F206 of HDAC2, and H172, F144, and F200 of HDAC3) along with the chelate Zn^2+^ were the key residues to stabilize the LIG2 in the catalytic active site. Although some residues fluctuated massively in some complexes than in others, the structural variations in the residues in the catalytic site were constant. Due to the H-bond and π-stacking interactions between the ligands and amino acids of the loop regions, this resulted in lower RMSF values in ligand-bound HDAC complexes. Residues in the loop regions were also engaged in the hydrogen bond interactions and chelation with Zn^2+^, resulting in stabilization. Conclusively, these mobile loop regions are observed with key stabilizing residues.

### 2.9. Post-Simulation Trajectories Analysis

The principal component analysis facilitates identifying the overall mobility of c-alpha atoms across structural ligand-bound complexes during the 100 ns of the simulation timeframe and evaluating the behavior of conformational changes using acquired principal components [40]. The major components of protein motions are isolated by employing principal component analysis (PCA). Changes in the dynamics of these fundamental motions of any macromolecule can cause protein malfunction or dysfunction. To evaluate protein mobility in ligands bound with HDAC complexes during the simulation, Cα- based principal component analysis was employed. The binding cluster components have been divided into green and red dots to coordinate groupings. These dots depict the process of flipping over conformational changes for each frame of the simulation system. The plot depicting the first few principal components (PCs) of the residues contains the greatest percentage of the entire mobility of the residues inside a dynamic system. As a result, PC1 and PC2 were computed using the MD simulation trajectories of all complexes. The two-dimensional patterns of HDACs movements within various inhibitor binding situations were identified by PCA, as shown in Figure 7. The PCA reveals the effects of LIG1 and LIG2 on HDAC conformational dynamics, demonstrating the impact of their binding in the catalytic site, i.e., the movable loop regions coordinated by Zn^2+^. The interactions of LIG1 and LIG2 induce positively correlated movements of residues engaged in binding. The PCA biplot among PC1 and PC2 of the LIG1 associated with HDAC1, 2, and 3 reveals the induced variability in enzymatic activity. However, the origin of PC1 of LIG1 bound with HDAC1, 2, and 3 resolves the array of conformational shifts, but it also reveals that the final cluster (red) is consolidated at maximum values of 29.24%, 41.86%, and 36.46%. This clearly shows that the HDACs structurally stabilize upon LIG1 binding in the catalytic active site. Nevertheless, during ligand interaction, the PCA of LIG2 also indicates a stable conformational analysis. Moreover, the red cluster aggregates with a smaller PC value of 15.41%, 37.33%, and 23.04%, indicating lower docking capability. This restricted mobility aligns with the findings of the Rg, which reveal that the introduction of LIG1 and LIG2 causes a contraction in the 3D conformation of HDAC. Such tightness in the protein structure effectively hinders its interactions with the substrate (Figure 7).

### 2.10. Free Energy Landscape of the Ligand-Bound Complexes

Free energy landscapes (FELs) serve an important role in the investigation of ligand binding via protein conformational equilibrium and stability. It provides useful information about the stability and metastability of protein–ligand complexes that may exhibit their dynamic behavior [41]. The FELs plot depicts the conformational states of ligand-bound HDAC complexes exhibiting lower energy represented in dark blue and shaded toward the global minimum. The FEL reveals the unique energy landscapes for both the ligands bound with HDACs enzyme, marginally altering the size and positions of the phases enclosed inside a stable global minimum. In particular, the energy landscape span from 0 to 7.87 kcal/mol for the LIG1-bound HDAC1, 0 to 8.88 kcal/mol for the HDAC2–LIG1 interaction followed by 0 to 10.03 kcal/mol for the LIG1-bound HDAC3 complex (Figure 8A–C). The FEL of the LIG1-bound HDAC2 complex is significantly similar to that of the bound HDAC3, indicating minor changes in conformational rearrangements. Similarly, for LIG2-mediated interactions, the energy ranges from 0 to 6.0 kcal/mol for HDAC1, 0 to 7.1 kcal/mol for HDAC2, and 0 to 8.0 kcal/mol for HDAC3 (Figure 8D–F). This characteristic emphasizes the investigation of several conformational changes in each system involving HDAC ligands. The LIG1-bound HDAC1 complex showed global minima with more energy basins than HDAC2 and HDAC3. Likewise, the LIG2-bound HDAC1 and HDAC2 complexes displayed single global minima with two–three local energy basins with different populations compared to the HDAC3 bound complex. Furthermore, LIG1-HDAC1, LIG2-HDAC1, and LIG2-HDAC2 were observed with two or more minima confined with two–three basins, demonstrating that there were numerous conformations towards obtaining the global minimum, as shown in Figure 8. Metastability in FELs revealed the presence of different binding modes or transitional conformations that the complex of protein and ligand may adopt throughout its dynamic trajectory. These metastable states might be transient intermediate structures that emerge during ligand binding, revealing important insights into the dynamics and kinetics of the ligand binding process. Interestingly, the inspection of free energy wells in the FELs demonstrated that the interaction of LIG1 and LIG2 with HDACs does not lead to protein unfolding and is structurally stable throughout the simulation. Thus, the PCA followed by FEL results support the dynamic studies of the complexes and suggest that the flexibility and structural variation decreases after the engagement of ligands in the binding pocket of HDAC enzymes.

### 2.11. Dynamic Cross-Correlation Matrix (DCCM)

To explore the functional displacements of ligand-bound HDAC complexes, we developed and examined a dynamic cross-correlation matrix (DCCM). The cross-correlations between the residues were detected by examining the trajectories of LIG1 and LIG2 bound with HDACs enzymes. The DCCM and PCA of the MD frames were computed by employing Bio3D R scripts to analyze the correlated movements of each complex. The DCCM generates a heat map of correlated residues, indicating whether the displacements of atoms in residues are positively and negatively correlated. There were variances in the DCCM graphs, owing to the structural and conformational shifts in the mobile loops of HDACs by LIG1 and LIG2 exclusively. The highly colored portions reveal a positive and negative correlation between the residues of target proteins, whereas the uncolored portions exhibit no such correlations. The highly correlated matrix denotes strong contact between the compounds and the HDACs binding pocket, leading to coordinated movements across the protein structure. The representations of DCCM from 0 to 100 ns of all complexes are shown in Figure 9. The zone in cyan color is strongly correlated, whereas the pink zone is significantly anti-correlated. As the color moves from cyan to white, the correlation decreases, whereas anti-correlation increases as the color moves from white to pink. The positively correlated residues are shown by the rectangular region. This allowed us to analyze the mechanism of variations in LIG1 and LIG2 bound with HDACs structures, and we found that certain areas have unique flexibility patterns. The results suggest that changes in the binding sites are minor but greatest in the loop regions.

### 2.12. Structural Deviations in HDACs

During course simulations (100 ns), the structural conformation of protein alterations overlaps between the initial (0 ns) and final (100 ns) conformations of HDACs, providing a clear picture of how many deviations are generated in the protein assembly. Upon ligand binding, HDACs have substantial RMSD deviations of 1.836 Å, 1.728 Å, and 1.262 Å with LIG1. On the other hand, the RMSD divergence of LIG2 bound with HDACs are 0.823 Å, 1.075 Å, and 0.677 Å, as depicted in Figure 10. Apart from the major differences in the loop areas across Zn^2+^, other areas such as α-helices and β-sheets also show notable changes in the structure.

### 2.13. Drug Likeness and BOILED-Egg Model

The blood–brain barrier (BBB) poses a significant challenge in therapeutic research, and there are currently no neurotherapeutic biological molecules that effectively cross the BBB for treating CNS disorders [42,43]. More than 98% of small molecules are prevented from entering the brain by the BBB [44]. Hence, it is important to consider the BBB transition activity of molecules from the beginning of therapeutic development for CNS diseases. One of the most important aspects of oral bioavailability pertaining to the intestinal system is the quantity absorbed by the drug candidates, and it is necessary to investigate GI absorption. P-Glycoprotein (PGP) is a part of the ABC transporter and ATP-binding cassette family (ABC) [45]. It functions as a biological shield, facilitating the removal of toxins and xenobiotics from cells while also playing a critical role in drug absorption and extraction processes [46]. Multi-drug resistance (MDR) is caused by the overexpression of ABC transporters, and it is one of the key reasons for treatment failures, such as cancer treatments [47]. By imputing the SMILES of LIG1, LIG2, and LIG3, the BOILED-Egg approach predicts human gastrointestinal absorption and blood–brain barrier permeability. The lipophilicity water/lipid octanol partition coefficient (WLOGP) and total polar surface area (TPSA) are two physiochemical factors used in the algorithm. The model was improved by using datasets of well-known and fewer known compounds in BBB and GI absorption [48]. In the model shown in Figure 11, the yellow area indicates that the molecule has good penetration and absorption properties in both the BBB and GI tract. We may infer that the placement of red dots for LIG1 and LIG2 showed negative properties for BBB penetration and PGP effect and positive properties for GI absorption, whereas TPSA is the key reason for negative BBB results. The TPSA of LIG1, LIG2, and LIG3 is 79.80 Å^2^, 95.00 Å^2^, and 52.66 Å^2^, which is outside of the acceptable domain region, as tabulated in Table 3. As shown in Table 3 and Table 4, LIG1, LIG2, and LIG3 exhibited favorable pharmacokinetic characteristics, encompassing drug likeness and pharmacological properties. All three compounds demonstrated sound physiochemical properties and efficient absorption within the gastrointestinal tract, although they were unable to traverse the blood–brain barrier. Furthermore, secondary metabolites have been evaluated for toxicity using the ProTox-II tool which classifies compounds into toxicity classes based on their lethal dosage (LD50). The class 1 and 2 compounds are lethal, class 3 is toxic, followed by class 4 and 5 compounds that are harmful. The class 6 compounds are nontoxic and suitable for human ingestion. The toxicity of LIG1, LIG2, and LIG3 was assessed using the ProTox-II online tool, and the values were presented in Table 4. Notably, LIG2 displayed moderate to low toxicity, while LIG1 and LIG3 exhibited higher LD50 values.

### 2.14. Prediction of Biological Activity

We then used the PASS online server evaluation to forecast the biological activities of chemical compounds. It examines both 2D and 3D descriptors to gain an understanding of the association between chemical structures and biological activity. The prediction analysis of compounds was provided as a ratio of probable activity (Pa) to probable inactivity (Pi). Pa and Pi values range from 0.000 to 1.000. In the present study, the top most significant activities with Pa > Pi and Pa > 1.000 were selected for further analysis and interpretation. The result revealed that LIG1, LIG2, and LIG3 had significantly predicted biological activities (PBAs), including three PBAs attaining Pa values of up to 0.9. All three compounds demonstrated several pharmacologically significant biological effects, including mycothiol-S-conjugate amidase inhibitor, peptide agonist, NADPH-cytochrome-c2 reductase inhibitor, neurotransmitter uptake inhibitor, complement factor D inhibitor, chloride peroxidase inhibitor, feruloyl esterase inhibitor, prolyl aminopeptidase inhibitor, and aspulvinone dimethylallyl transferase inhibitor (as shown in Table 5).

## 3. Materials and Methods

### 3.1. Selection and Refinement of HDACs

The structures of HDACs enzymes were retrieved from the Protein Data Bank with [PDB IDs: 6Z2K-HDAC1, 7LTG-HDAC2, 4A69-HDAC3 (http://www.rcsb.org, accessed on 5 October 2021) [49]. The pose validation approach is widely used in the various methods for validating docking algorithms [50]. The crystal structure of HDAC2 bound with co-crystallized ligand apicidin was redocked in the catalytic site to find an exact confirmation of Zn^2+^ chelation. Furthermore, Molecular Operating Environment (MOE) was utilized to prepare the structures prior to molecular docking.

### 3.2. Ligand Database Preparation

The ZINC database is a public database containing by a million compounds for screening purposes (http://zinc.docking.org) [51]. The structures of ligands are available in both 2D and 3D formats. The library of drug-like molecules was retrieved from a ZINC database. All compounds were optimized using the MOE program by employing a 3D protonate module for adding partial charges. The MMFF94X forcefield was used for the energy minimization of all molecules and was added to the MOE ligand database for molecular docking [52].

### 3.3. Virtual Screening and Molecular Docking

Molecular docking was utilized for screening a drug-like library of 5000 compounds employing MOE software. The extensive literature was examined to confirm the catalytic active site of HDAC proteins. Likewise, the site finder tool of MOE [53] was utilized to investigate the key player binding residues of HDACs with Zn^2+^ and generate electrostatic surface maps that include these residues to identify docking active sites. The drug-like library of compounds along with co-crystalized ligand apicidin were subjected to molecular docking into the specified docking sites of HDACs enzymes [54]. We applied triangular algorithms to identify the ten different docking poses of each molecule. London dg function was used to rescore the simulated poses and to generate the top 10 different poses as per molecules, which were then minimized using force fields refinements algorithm, and binding energy was computed using generalized Born solvation models while keeping the receptor residues rigid [55]. Moreover, the top eight compounds were ranked as per binding energy, S-score function, and root mean square deviations (RMSDs). In addition, the top poses were selected for further study based on their energy and binding interactions with the catalytic active site of HDAC enzymes.

### 3.4. Ligand Receptor Interaction Analysis

The MOE LigX tool was employed to create two-dimensional patterns of protein–ligand binding interactions, highlighting hydrogen bonds, electrostatic/nonelectrostatic interactions, and hydrophobic interactions for ligand–receptor interaction analysis [56]. The interaction mechanism of the ligands helps in validating firm binding inside the catalytic pocket of HDAC enzymes. In addition, PyMOL software was employed to construct the three-dimensional structures of protein inhibitor-bound complexes [57]. The co-crystalized ligand and library compounds were docked to the HDAC enzymes after implementing the “Lipinski’s Rule of Five” to the compounds using scan tools at the SWISSADME server [58]. The top three hits’ binders selected from molecular docking based on docking scores and the interaction mode against all HDAC enzymes were subjected to molecular dynamic simulation studies.

### 3.5. Molecular Dynamics Simulation

Molecular dynamics (MD) simulations were conducted for LIG1 and LIG2, which were identified through virtual screening analyses targeting selective HDACs. These simulations were performed using GROMACS v5.1.4 (http://gromacs.org) over a 100 ns timeframe to assess the stability of the ligand–protein complexes. The complexes were placed in a cubical box, ensuring a minimum distance of 10 Å between any ligand atom and the box’s edges. The GROMOS96 53a6 forcefield was employed to prepare these ligand–protein complexes using the TIP3 water model, and Na+/Cl− ions were added to neutralize the system and balance the charges [59]. The initial energy minimization of the system involved 5000 steps using the steepest descent algorithm, with a convergence threshold set to less than 1000 kcal/mol/nm. After completing the initial minimization, the entire system underwent equilibration for 5 ns at 300 K and 1 bar pressure, utilizing both the canonical (NVT) and isothermal isobaric (NPT) ensembles. The thermostat coupling was set at a reference temperature of 300 K using a Berendsen thermostat, while pressure coupling was maintained at a reference pressure of 1.0 bar, employing Parrinello–Rahman along with periodic boundary conditions that included cut-offs for Lennard–Jones and Coulomb interactions [60]. Long-range interactions were handled using the Particle Mesh Ewald (PME) algorithm. Subsequently, the final MD simulation was conducted for LIG1 and LIG2 ligand–protein complexes over a 100 ns timeframe. A time step of 2 fs was utilized, with coordinate data being saved at intervals of 10 fs. This methodology was employed to generate the root mean square deviation (RMSD) and root mean square fluctuation (RMSF) data [61].

### 3.6. MD Trajectory Analysis

Analyzing the MD trajectories of enzyme–drug complexes, the R studio (Bio3D package) was used to obtain information regarding cross-correlation across protein residues [62]. The residual information was visualized using the dynamical cross-correlation matrix and principal component analysis (PCA) followed by free energy landscapes to explore atomic mobility and stability [57,63]. PCA has proven a beneficial tool to expose protein’s principal movements to investigate folding dynamics. Additionally, the FELs of the ligand-bound HDAC complexes were created using the conformational sampling method to investigate the conformational variation.

The FELs graphs were generated using the following formula:ΔG(X) = −*K_B_T*ln *P*(*X*)
where *T* is the simulation’s temperature, *K_B_* denotes the Boltzmann constant, and *P(X)* denotes the probability distribution and the PCs.

Furthermore, the magnitude of DCCM’s correlation coefficients depends on the degree to which the system’s fluctuations are linked. Defining the correlation between the motions of distinct adjacent or distinct domains is one of the most important properties of the DCCM matrix. PCA can reflect the association between the considerable, complete motions recorded from the protein dynamics trajectories, allowing it to visualize variances in the dataset.

### 3.7. Prediction of Pharmacokinetics, Drug Likeness, and Physiochemical Properties

The retrieved hit compounds from virtual screening were subsequently selected to calculate their physiochemical, drug likeness, and pharmacokinetic and therapeutic properties. SwissADME (http://www.swissadme.ch) followed by the PASS online server were employed for predicting various properties (absorption, distribution, excretion, toxicity, drug likeness, and physiochemical properties) and other biological activities of the selected hit compounds [64,65].

### 3.8. Biological Activity Predictions

The biological properties of the hit compounds were evaluated using the PASS (prediction of activity spectra for substances) webserver [66]. This server examines compounds biological properties based on the structure activity. It provides a brief list of the potential biological properties based on the Pa ratio (probability of activity) and Pi ratio (probability of inactivity). A greater Pa value indicates a higher probability of biological activity for the hit compounds being studied.

## 4. Conclusions

This research delved into the development of novel potential cancer drugs targeting class 1 histone deacetylases (HDACs) and highlights the promise of two specific ligands, LIG1 and LIG2, as potent HDAC inhibitors with potential applications in cervical cancer treatment. Through interaction studies within protein–ligand complexes, it was evident that LIG1 and LIG2 chelating with the Zn^2+^ ions present in HDACs. This interaction is a crucial aspect of their potential as inhibitors. The findings were encouraging, with the top ligands showing low toxicity and minimal adverse effects on various physiological systems, including the blood, cardiovascular system, gastrointestinal system, kidney, liver, and lungs. Furthermore, these ligands exhibited favorable characteristics such as good passive absorption and high oral bioavailability. The results from screening, incorporating docking, dynamics, and drug scan studies, conclusively identified LIG1-2 as a potent inhibitor for HDAC1, HDAC2, and HDAC3. These findings underscore the importance of computational techniques in drug discovery, enhancing the efficiency of potential HDACIs. This strategic approach not only reduces the quantity of chemicals required in synthesis and biological testing but also significantly minimizes resource demands compared to conventional procedures. To further assess the potential of LIG1 and LIG2, molecular dynamic simulations were employed to predict the stability of the protein–ligand complexes. These simulations unveiled robust interactions and coordination with the Zn^2+^ ions of HDACs, signifying a high level of stability. The current findings proposed innovative cancer drug discovery targeting class 1 HDACs, highlighting LIG1 and LIG2 as potential inhibitors for cancer treatment, emphasizing their interaction pattern and stability in bounded complexes. However, further investigation in vitro and in vivo will be needed to validate their functional activity.

## Figures and Tables

**Figure 1 pharmaceuticals-17-00444-f001:**
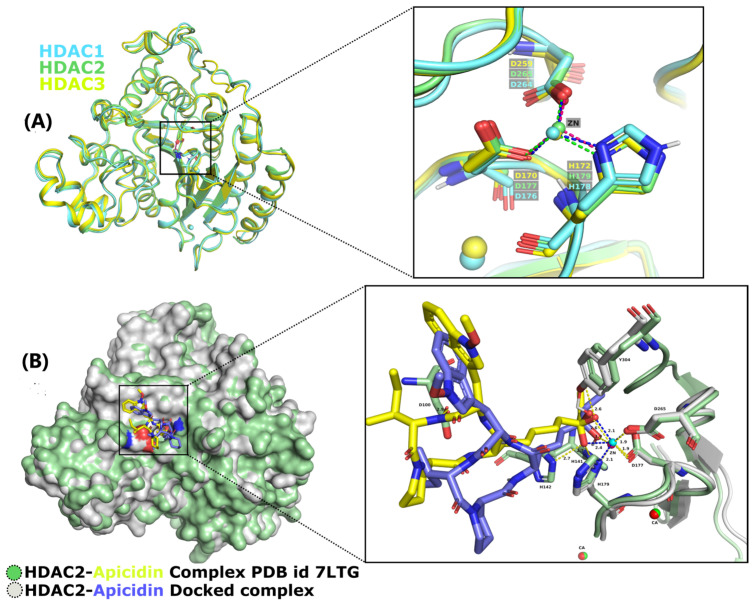
Structures of selective HDACIs. (**A**) Superimposition of HDACs’ catalytic site residues stabilize Zn^2+^ via metal coordinating bonds. (**B**) Surface mapping and superimposition of HDAC2 bound with crystalized structure and docked structure. In both panels, the residues within the catalytic site of HDACs were depicted in sticks. The solved crystal complex (7LTG) and docked apicidin in this work were colored yellow and blue, respectively. Residues in the active site D100, H141, H142, D177, H179, D265, and Y304 were also labeled. The Zn^2+^ ions are represented as spheres in the catalytic site of each complex that are involved in the coordination with apicidin and binding residues. Moreover, in both the panels, the hydrogen bonds and metal coordination bonds were shown as yellow, blue, magenta, and green dashes, respectively.

**Figure 2 pharmaceuticals-17-00444-f002:**
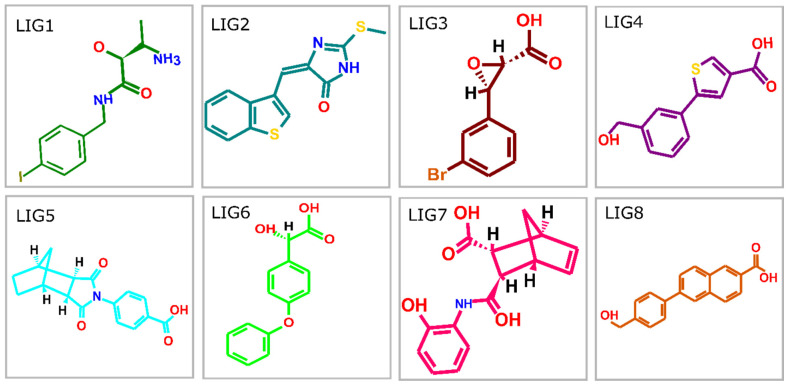
Chemical structures of the hit molecules based on binding S-score and interaction pattern.

**Figure 3 pharmaceuticals-17-00444-f003:**
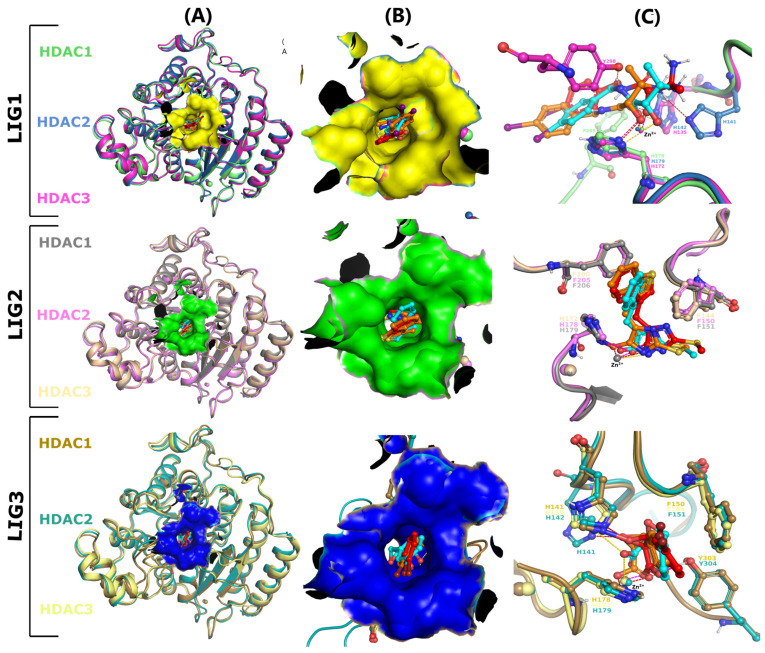
Structures of selective HDACs in complex with LIG1 (red), LIG2 (orange), and LIG3 (cyan). (**A**) Superimposition of HDAC1, 2, and 3 structures bound with LIG1, LIG2, and LIG3 in the catalytic site; (**B**) hydrophobicity surface showing that the inhibitors are bound deep in the hydrophobic pocket; (**C**) superimposition of HDAC1, 2, and 3 structures shows the interaction pattern bound with LIG1, LIG2, and LIG3. In all three panels, the residues and inhibitors at the hydrophobic active pocket were shown as ball and stick models. The Zn^2+^ ions at the active site of HDAC1, 2, and 3 that are involved in hydrogen bonding were shown as spheres. Coordination bonds and hydrogen bonds LIG1, LIG2, and LIG3 with their respective HDAC were shown as red, magenta, and yellow dashed lines, respectively.

**Figure 4 pharmaceuticals-17-00444-f004:**
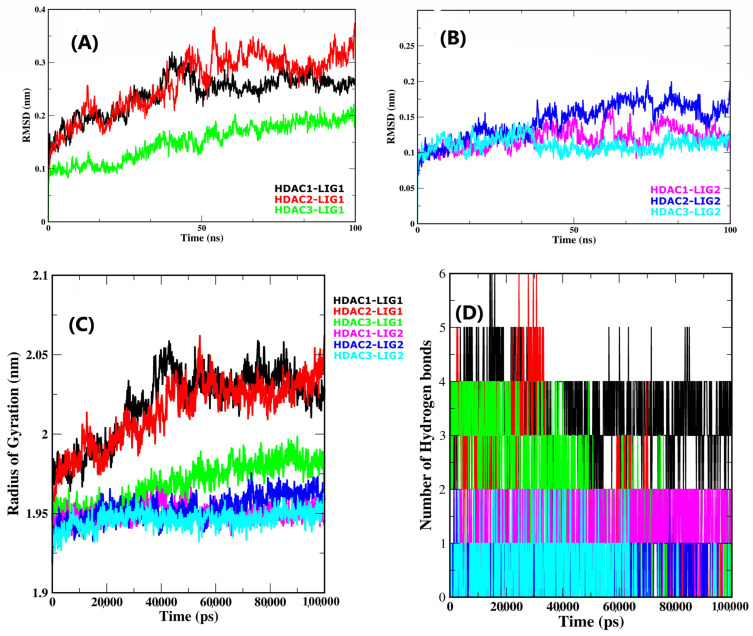
Comparison changes in RMSD values, Rg values, and intermolecular H bonds; (**A**) Rmsd deviations of LIG1-bound HDACs complexes, (**B**) Rmsd deviations of LIG2-bound HDACs complex, (**C**) Rg compactness of LIG1- and LIG2-bound HDACs complexes, (**D**) hydrogen bonds analysis of LIG1- and LIG2-bound HDACs complexes.

**Figure 5 pharmaceuticals-17-00444-f005:**
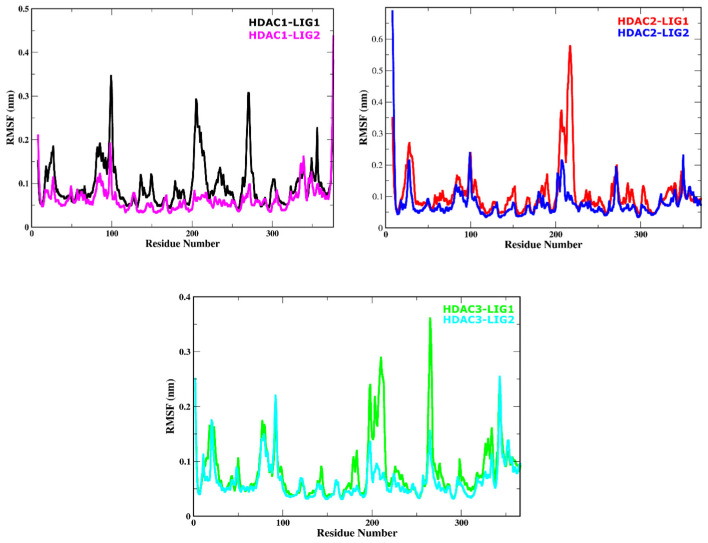
Root mean square fluctuation (RMSF) values of solvated HDAC enzymes bound with LIG1 and LIG2 were plotted versus residue number.

**Figure 6 pharmaceuticals-17-00444-f006:**
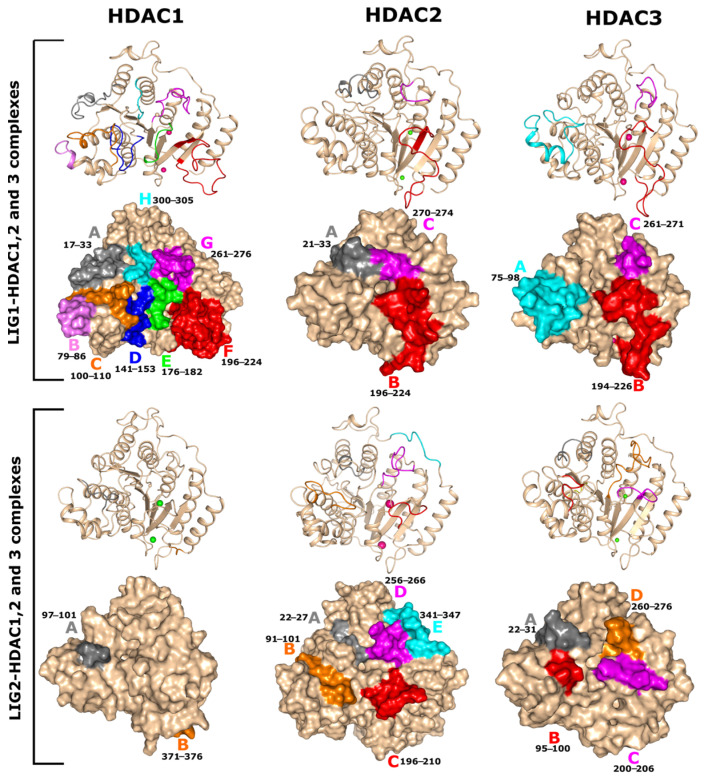
Structural mobility of and conformational changes in HDACs enzymes during 100 ns of the simulation system.

**Figure 7 pharmaceuticals-17-00444-f007:**
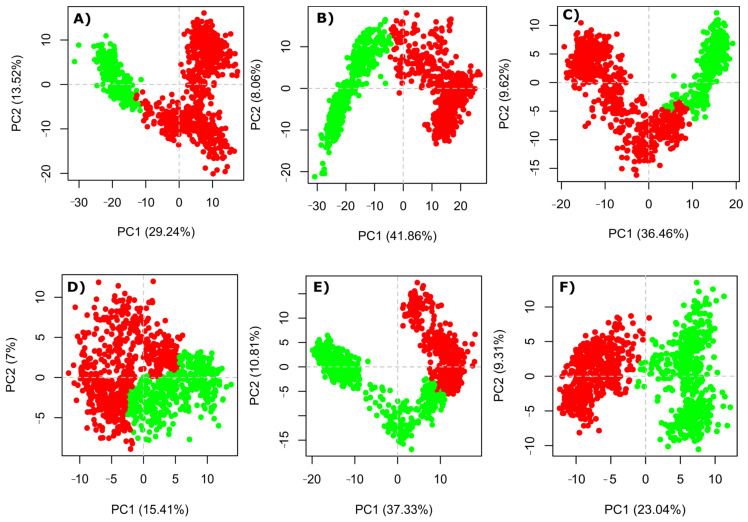
Comparison changes in PCA pattern of HDACs interactions with different LIG1 and LIG2; (**A**) LIG1−HDAC1 complex, (**B**) LIG1−HDAC2 complex, (**C**) LIG1−HDAC3 complex, (**D**) LIG2− HDAC1 complex, (**E**) LIG2−HDAC2 complex, and (**F**) LIG2−HDAC3 complex.

**Figure 8 pharmaceuticals-17-00444-f008:**
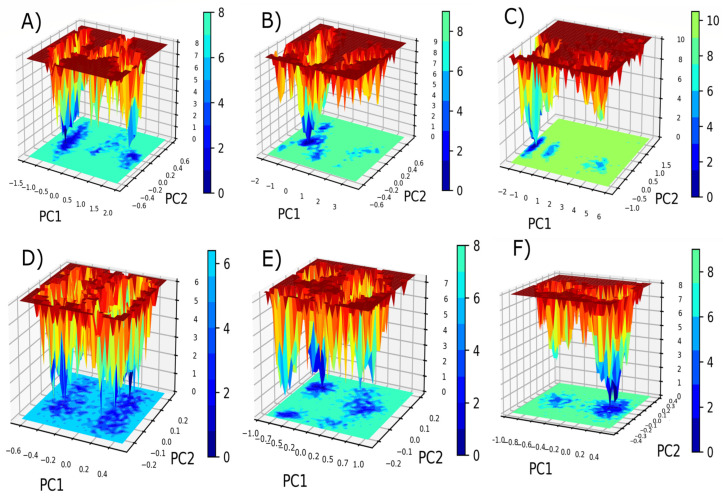
The FEL plots for the ligand-bound HDACs complexes. (**A**) LIG1-HDAC1 complex, (**B**) LIG1-HDAC2 complex, (**C**) LIG1-HDAC3 complex, (**D**) LIG2-HDAC1 complex, (**E**) LIG2-HDAC2 complex, and (**F**) LIG2-HDAC3 complex.

**Figure 9 pharmaceuticals-17-00444-f009:**
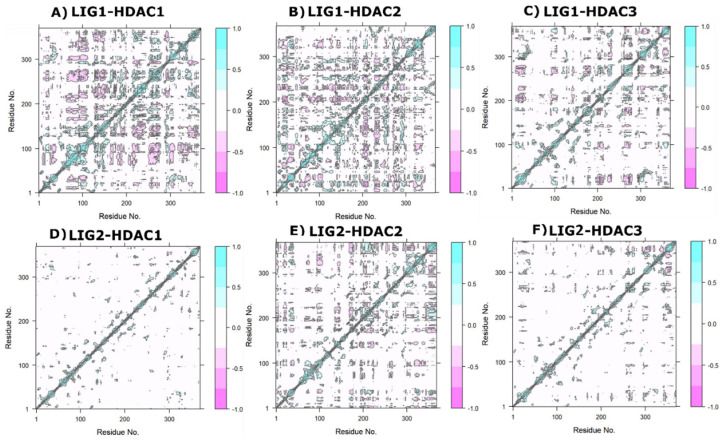
Plots of HDACs dynamic cross-correlation. (**A**) LIG1-HDAC1 complex, (**B**) LIG1-HDAC2 complex, (**C**) LIG1-HDAC3 complex, (**D**) LIG2-HDAC1 complex, (**E**) LIG2-HDAC2 complex, and (**F**) LIG2-HDAC3 complex. The coloring changes from pink (−), white (0), and cyan (+). The negative value shows anti-correlation, which means the atoms are moving in the opposite direction, while the positive value shows correlated mobility, which means atoms are moving in the same direction.

**Figure 10 pharmaceuticals-17-00444-f010:**
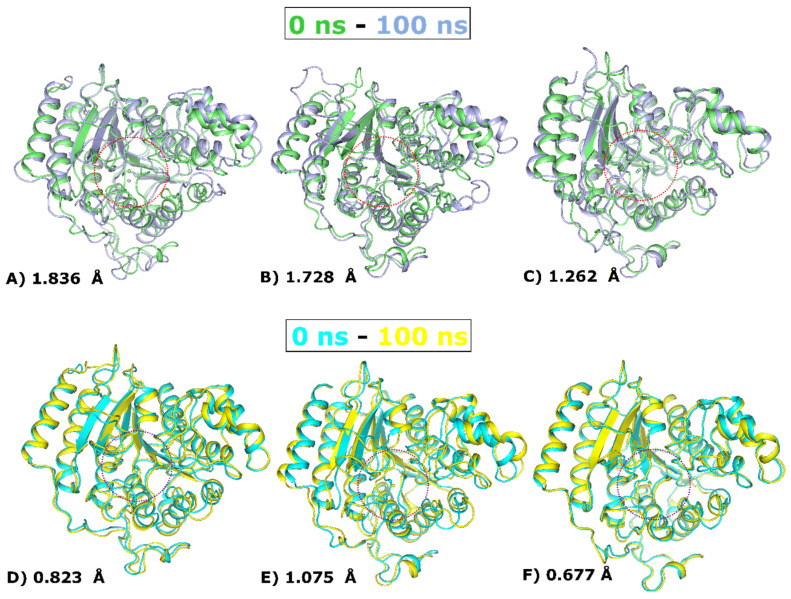
Structural imposition at 0 ns and 100 ns timeframe. (**A**) LIG1-HDAC1, (**B**) LIG1-HDAC2, (**C**) LIG1-HDAC3, (**D**) LIG2-HDAC1, (**E**) LIG2-HDAC2, and (**F**) LIG2-HDAC3. The dotted circle represents the ligand binding site.

**Figure 11 pharmaceuticals-17-00444-f011:**
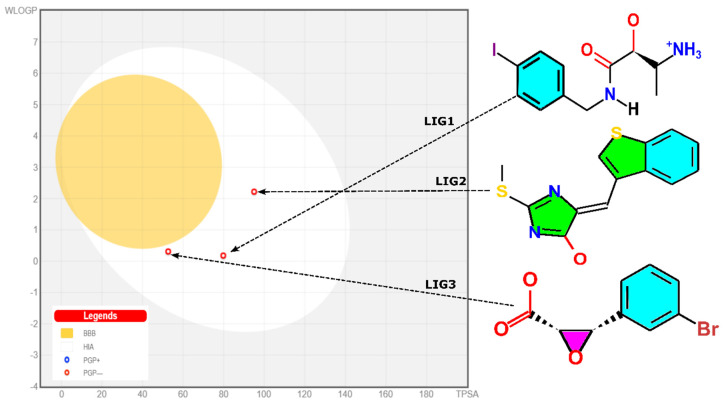
Evaluation of ligands’ permeability through the gastrointestinal tract and brain by BOILED-Egg method.

**Table 1 pharmaceuticals-17-00444-t001:** The top eight hit binders with their zinc ID were screened by MOE along with their S-score.

Compounds	S-Score (kcal/mol)		
	HDAC1	HDAC2	HDAC3
ZINC98207834-LIG1	−19.1103	−27.0314	−27.4039
ZINC77024375-LIG2	−21.8915	−23.2849	−27.9976
ZINC67801495-LIG3	−17.5725	−20.1273	−16.1137
ZINC85664535-LIG4	−17.1144	−18.8858	−15.1259
ZINC71792267-LIG5	−14.8882	−17.8317	−16.4172
ZINC71784493-LIG6	−14.6239	−17.4523	−16.7444
ZINC71792707-LIG7	−16.7516	−17.9025	−197871
ZINC79387365-LIG8	−15.3222	−16.7737	−15.7455

**Table 2 pharmaceuticals-17-00444-t002:** The binding interaction mechanisms of the top three hits inhibitors bound with the HDACs targets.

Complexes	Hydrogen Bond	π-Stacking Interactions	Hydrophobic Interactions
LIG1-HDAC1	H178	F205	D99, L139, H140, H141, G149, F150, C151, G300, G301, Y303
LIG2-HDAC1	H178	F150, F205	M30, D99, H141, G149, L271, G301, Y303
LIG3-HDAC1	H141, H178, Y303	F150	H140, G149, F205, L271, G301,
LIG1-HDAC2	H141, H142, H179	-	G150, F251, Y205, F206, L272, Y304, G302
LIG2-HDAC2	H179	F151 and F206	M31, D100, L140, H141, H142, G150, C152, G301, G302, Y304,
LIG3-HDAC2	H141, H142, H179, Y304	F151	D100, G150, F206, L272, G302
LIG1-HDAC3	H135, H172, Y298	-	M24, H134, G143, F144, F200, L266, G295, G296
LIG2-HDAC3	H172	F144 and F200	M24, D93, H134, H135, G143, C145, L266, G296, Y298
LIG3-HDAC3	H135, H172	F144	D93, H134, G143, C145, F200, L266, G296, Y298

**Table 3 pharmaceuticals-17-00444-t003:** Drug likeliness of compounds (SwissADME).

Compounds	Lipinski	Ghose	Veber	Egan	Muegge	Bioavailability Score
LIG1	Yes	Yes	Yes	Yes	Yes	0.55
LIG2	Yes	Yes	Yes	Yes	Yes	0.55
LIG3	Yes	No	Yes	Yes	Yes	0.85

**Table 4 pharmaceuticals-17-00444-t004:** Pharmacological properties and toxicity prediction results for top compounds.

Compounds	GI Absorption	BBB Permeation	P-Glycoprotein Substrate	CYP1A2 Inhibitor	CYP2C19 Inhibitor	CYP2C9 Inhibitor	CYP2D6 Inhibitor	CYP3A4 Inhibitor	LD 50 mg/kg	Carcinogenicity, Mutagenicity
LIG1	high	no	no	no	no	no	no	no	3500	probably safe
LIG2	high	no	no	yes	yes	yes	no	yes	800	probably safe
LIG3	high	no	no	no	no	no	no	no	2300	probably safe

**Table 5 pharmaceuticals-17-00444-t005:** The predicted biological activities of LIG1, LIG2, and LIG3 using the PASS online server.

Compounds	Pa	Pi	PBA
LIG1	0.925	0.001	Mycothiol-S-conjugate amidase inhibitor
	0.837	0.004	Peptide agonist
	0.762	0.010	NADPH-cytochrome-c2 reductase inhibitor
LIG2	0.603	0.029	Neurotransmitter uptake inhibitor
	0.508	0.068	Complement factor D inhibitor
	0.495	0.055	Chloride peroxidase inhibitor
LIG3	0.914	0.004	Feruloyl esterase inhibitor
	0.904	0.004	Prolyl aminopeptidase inhibitor
	0.868	0.015	Aspulvinone dimethylallyl transferase inhibitor

## Data Availability

Data contained within the article and Appendix A.

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
