# Peer review of "Investigating Potential Cancer Therapeutics: Insight into Histone Deacetylases (HDACs) Inhibitions"

_pharmaceuticals, 2024, doi:10.3390/ph17040444_

Round 1

Reviewer 1 Report

Comments and Suggestions for Authors

1. The methods need to include quantitative details as well as citations to original work; not just the programs used. In order to make the work reproducible, capable of being evaluated, and to provide credit to those who developed the methods used in the manuscript. like the solvent model, the thermostat, barostat and molecular dynamics details.

2.Are the molecular dynamics results consistent with the molecular docking results? Whether the selected conformations are the same, please reflect in the manuscript

3.Observed molecular docking results of hydrogen bonding Table 2, and molecular dynamics simulation results of hydrogen bonding Fgure 3(D) , Please explain the reasons.

4.The results of solvent-accessible surface area (SASA)analysis were supplemented.

5.Whether HDAC2-LIG1 is stable at 100ns in Figure 3 (A) cannot be judged, so please increase the simulation time to 120ns or 150ns to further determine whether it is stable.

6. Please analyze the conformation of the lowest energy point of PCA analysis specifically and compare it with the conformation of the molecular docking results.

7.Supplementary Molecular dynamics MMGBSA free energy calculation and free energy decomposition data.

Author Response

I wanted to express my sincere gratitude for taking the time to review our manuscript. Your insightful comments and suggestions have been immensely valuable in enhancing the quality and clarity of our work. I am pleased to inform you that we have diligently addressed all the major comments provided by you and have incorporated them into the revised version of the manuscript. Additionally, we have prepared a detailed response document outlining the changes made and our rationale behind them. I have attached the response file for your perusal. Once again, thank you for your invaluable feedback and for your commitment to scholarly excellence.

REVIEWER 1

Comment 1. The methods need to include quantitative details as well as citations to original work; not just the programs used. In order to make the work reproducible, capable of being evaluated, and to provide credit to those who developed the methods used in the manuscript. like the solvent model, the thermostat, barostat and molecular dynamics details.

Response.  Thank you for your insightful comments and suggestions regarding our manuscript. In response to your feedback, we will enrich the manuscript by including more comprehensive details about the solvent model, thermostat, and barostat settings used in our molecular dynamics simulations of LIG1 and LIG2. We recognize the necessity of these details for reproducibility and proper credit to the original methodologies. Additionally, we will provide a clearer explanation for our choice of the Berendsen thermostat and Parrinello-Rahman barostat, and their specific parameters. We agree that these enhancements will not only improve the clarity of our methods but also offer a more profound understanding to t he readers.

Comment 2. Are the molecular dynamics results consistent with the molecular docking results? Whether the selected conformations are the same, please reflect in the manuscript.

Response. 

Thank you for highlighting the need to compare molecular dynamics and docking results in our study. We analyzed the 100 ns MD simulation data to see if the ligand conformations align with our docking predictions, and updated our manuscript to reflect these findings, enhancing its scientific rigor.

Comment 3. Observed molecular docking results of hydrogen bonding Table 2, and molecular dynamics simulation results of hydrogen bonding Fgure 3(D) , Please explain the reasons.

Response.  Thank you for the valuable comment. Protein-ligand interactions and MD simulations provide valuable information about hydrogen bonds, they serve different purposes and offer complementary insights into the behavior of these interactions in biological systems. The differences between observed hydrogen bonding patterns in molecular docking results and MD simulation graphs are due to the inherent limits of static docking studies and the dynamic nature of MD simulations. Moreover, molecular docking sheds light on potential hydrogen bonding interactions at a single conformation, but MD simulations provide a more comprehensive perspective of hydrogen bonding dynamics over time, capturing the complex dynamics of molecular interactions in a biological system. Thus Table 2 represents the interaction pattern and formation of hydrogen bonds mentioned at a single conformation while the plot of hydrogen bonds in the MD section represents the hydrogen bond formation over time.

Comment 4. The results of solvent-accessible surface area (SASA) analysis were supplemented.

Response.  Thank you for your valuable comment. The results of the SASA analysis were incorporated in the manuscript and the plots were added as supplementary figure.

Comment 5. Whether HDAC2-LIG1 is stable at 100ns in Figure 3 (A) cannot be judged, so please increase the simulation time to 120ns or 150ns to further determine whether it is stable.

Response.  Thank you for your comment. In this investigation, we conducted 100 nanosecond simulations for both LIG1 and LIG2 in complex with each of the HDAC1, HDAC2, and HDAC3 proteins. Comparative analyses were performed for each complex, examining the behavior of the ligands within the binding pocket. Our findings indicate that LIG2 exhibits greater stability compared to LIG1 across all HDAC enzymes. Additionally, trajectory analyses of HDAC3 revealed significant results in comparison to HDAC1 and HDAC2. Therefore, we conclude that LIG2 demonstrates superior stabilization relative to LIG1 when binding to HDAC enzymes. Furthermore, LIG1 in complex with HDAC2 displays heightened fluctuations compared to interactions with other enzymes. As a result, the 100 ns of simulation time allowed sufficient to bring the systems under investigation to equilibrium for comparative analysis.

Comment 6. Please analyze the conformation of the lowest energy point of PCA analysis specifically and compare it with the conformation of the molecular docking results.

Response.  We acknowledge the request for an analysis of the conformation of the lowest energy point from PCA analysis and its comparison with the molecular docking results. We would like to highlight that we have conducted a comprehensive free energy landscape analysis to identify the lowest energy point and incorporate in the revised manuscript, which provides a robust representation of the stability. This approach allows us to capture the dynamics and stability of the protein-ligand complex more effectively compared to a singular snapshot obtained from molecular docking. Moreover, our study incorporates both PCA analysis and free energy landscape techniques to elucidate the structural dynamics and stability of the protein-ligand complex, offering a more comprehensive understanding of the molecular interactions under investigation. We believe this approach enhances the reliability and validity of our findings.

Comment 7. Supplementary Molecular dynamics MMGBSA free energy calculation and free energy decomposition data

Response.  Thank you for your suggestion regarding the use of MMPBSA for free energy calculations in our study on HDAC1, 2, and 3. We acknowledge the importance of such methods in biomolecular research. However, we face a specific challenge with HDACs, which are zinc-containing metalloenzymes. The standard MMPBSA approach may not accurately represent the metal-ligand interactions essential for these enzymes, as typical force fields used in these calculations often fall short in modeling the complex electronic nature of metal ions like Zn2+. Given this limitation, we are cautious about the reliability of MMPBSA results for our HDAC-focused study. We are exploring alternative computational approaches that can better accommodate the unique characteristics of metal ions in these enzymes.

Reviewer 2 Report

Comments and Suggestions for Authors In the current work, an international group of researchers identified novel selective inhibitors of HDACs using a complex in silico approach, including high-throughput molecular docking, molecular dynamics and prediction of key pharmacokinetic properties of hit compounds. Ahmad and colleagues identified three compounds (LIG1, LIG2, LIG3) from the ZINC database that showed the lowest S-score for all three types of HDACs and, using MD simulations, verified the direct interaction of LIG1 and LIG2 with the active site of the studied proteins under dynamic conditions. This work is characterized by scientific novelty, arouses great interest and can be published in a special issue "Computer-Aided Drug Design and Drug Discovery" of the Pharmaceuticals, but after some revisions.   Major comments: 1. Why were molecular dynamics calculations performed for only two hits, LIG1 and LIG2 (Sections 3.2-3.8)? Why was LIG3 ignored? Please provide a detailed response and make corrections to the text. 2. since LIG1 carries a charged group -NH3+, please comment on the contribution of electrostatic interaction of LIG1 with amino acid residues to the stabilization of ligand-protein complexes LIG1-HDACs based on molecular modeling data? 3. Lines 453-454 - Please describe the data stratification for PCA analysis in more detail and clarity. It is unclear from the text into which two groups you stratified the analyzed samples? 4. To obtain more objective results for the prediction of pharmacokinetic properties, you should use several chemoinformatics resources based on different algorithms (not only SwissADME). Try to complement your results for this block with analyses on other web platforms, e.g. services of PASS Online (https://www.way2drug.com/passonline/services.php). 5. Line 560 - Are you sure that the analyzed ligands form covalent bonds with Zn2+? Please comment or correct.   Minor comments: 1. Please give the chemical structures of the studied molecules LIG1, LIG2, LIG3 in section 3.1 (in the current version of the manuscript these structures are given at the end of the paper in Fig. 9). 2. line 72 - Zn 2+ - please specify 2+ as degree 3. line 77 - please decipher the abbreviation HDACIs. The abbreviation HDACIs is only given in line 82. 4. lines 92, 111 - "HDACIs inhibitors" means "HDAC inhibitors inhibitors". Please correct. 5. line 104 - what does HDACI mean? Perhaps you mean HDACIs? 6. line 131 - please capitalize zinc 7. line 201 - please replace SwissSME with SwissADME 8. Table 2 - please create separate columns for hydrogen bonding and pi-stacking interactions 9. Figure 3 - please add labels for graphs (A, B, C, D) 10. Line 357 - the phrase "enzyme HDAC's" is incorrect. Please correct this. 11. Line 412 - please capitalize "Only". 12. Please swap Figure 6 and Figure 7. 13. Line 530 - please capitalize "smiles". 14. Lines 576-577 - please correct "invitro and invivo". Comments on the Quality of English Language

The authors need to reread the manuscript several times because there are a significant number of missing spaces, capitalized words in the middle of sentences, and the atomic charge is not given in terms of degrees.

Author Response

I wanted to express my sincere gratitude for taking the time to review our manuscript. Your insightful comments and suggestions have been immensely valuable in enhancing the quality and clarity of our work. I am pleased to inform you that we have diligently addressed all the major and minor comments provided by you and have incorporated them into the revised version of the manuscript. Additionally, we have prepared a detailed response document outlining the changes made and our rationale behind them. I have attached the response file for your perusal. Once again, thank you for your invaluable feedback and for your commitment to scholarly excellence.

REVIEWER 2

Major Comments

Comment 1: Why were molecular dynamics calculations performed for only two hits, LIG1 and LIG2 (Sections 3.2-3.8)? Why was LIG3 ignored? Please provide a detailed response and make corrections to the text.

Response: Thank you for your observations and comments. The choice to carry out MD simulation for just two hits LIG1 and LIG2 was made considering the binding mode inside the active site of HDAC enzymes and their binding scores. While LIG3 demonstrated outstanding binding scores and interaction patterns during docking analysis, further MD simulations indicated that LIG3 could not achieve any stability and worthwhile findings as LIG1 and LIG2. The instability found in LIG3 during the simulations motivated us to concentrate our extensive investigation on LIG1 and LIG2, which demonstrated consistent and positive behavior throughout the simulation. This selective strategy enabled us to identify and probe deeper into the ligands with the most promising properties for investigation.

Comment 2. Since LIG1 carries a charged group -NH3+, please comment on the contribution of the electrostatic interaction of LIG1 with amino acid residues to the stabilization of ligand-protein complexes LIG1-HDACs based on molecular modeling data?

Response:  Thank you for highlighting the charged nature that exists in the -NH3+ moiety in LIG1 and its potential function in stabilizing the ligand-protein complexes with HDACs. In our molecular docking data, we discovered a surprising interaction involving the -NH3+ group of LIG1 with residue H141 in the binding site of HDAC2. A hydrogen bond is formed during this contact between one hydrogen of the -NH3+ and H141 as mentioned in the manuscript and also here in response (2D interaction Figure 1). Hydrogen bonding seems an influential factor in identifying molecules and the specific interaction of NH3+ and H141 is crucial to the stability of the LIG1- HDAC2 complex. On the other hand, the absence of interaction with other HDAC isoforms changes in amino acid composition that impairs electrostatic complementarity towards the charged group of LIG1.

Figure 1 : 2D interaction pattern of LIG1 and HDAC2 complex.

Comment 3:  Lines 453-454 - Please describe the data stratification for PCA analysis in more detail and clarity. It is unclear from the text into which two groups you stratified the analyzed samples?

Response: Thank you for highlighting.  The PCA analysis was carried out by categorizing the MD simulation trajectories into discrete groups reflecting various binding scenarios. There are six complexes in total and divided into three groups with HDAC isoforms. Similarly, the first and second principal components (PC1 and PC2) were computed for each group to identify and show the primary components of protein movements in each binding scenario.

Comment 4:  To obtain more objective results for the prediction of pharmacokinetic properties, you should use several chemoinformatics resources based on different algorithms (not only SwissADME). Try to complement your results for this block with analyses on other web platforms, e.g. services of PASS Online (https://www.way2drug.com/passonline/services.php).

Response: Thank you for the valuable comment. The prediction of pharmacokinetic properties was analyzed by employing an additional PASS Online server as you suggested. The data is incorporated in the manuscript as per your comment.

Comment 5. Line 560 - Are you sure that the analyzed ligands form covalent bonds with Zn2+? Please comment or correct.

Response: Thank you for the comment. The interaction pattern revealed the chelation of Zn2+ ion with LIG1 and LIG2, stabilized by the residues of the HDAC isoforms in the active site. Similarly, both the ligands LIG1 and LIG2 show significant interaction with Zn2+ ion as mentioned in the 3D interaction Figure. 

Minor Comments

Comment 1. Please give the chemical structures of the studied molecules LIG1, LIG2, LIG3 in section 3.1 (in the current version of the manuscript these structures are given at the end of the paper in Fig. 9).

Response: Thank you for the comment. The chemical structures of all the ligands are incorporated in the manuscript.

Comment 2.. line 72 - Zn 2+ - please specify 2+ as the degree

Response: Thank you for the comment. Zn 2+ is specified as degree in the whole manuscript.

Comment 3. line 77 - please decipher the abbreviation HDACIs. The abbreviation HDACIs is only given in line 82.

Response: Thank you for the comment. The abbreviation of HDACIs is incorporated in the manuscript.

Comment 4. lines 92, 111 - "HDACIs inhibitors" means "HDAC inhibitors inhibitors". Please correct.

Response: Corrected

Comment 5. line 104 - what does HDACI mean? Perhaps you mean HDACIs?

Response: Histone deacetylase inhibitors is the full abbreviation of HDACI.

Comment 6. line 131 - please capitalize zinc

Response: Corrected

Comment 7. line 201 - please replace SwissSME with SwissADME 8

Response: Corrected

Comment 8. Table 2 - please create separate columns for hydrogen bonding and pi-stacking interactions. 

Response:  Thank you for the comment. The columns for h-bonds and pi stacking interactions are separated .

Comment 9. Figure 3 - please add labels for graphs (A, B, C, D) 

Response: Corrected

Comment 10. Line 357 - the phrase "enzyme HDAC's" is incorrect. Please correct this.

Response: Corrected

Comment 11. Line 412 - please capitalize "Only". 

Response: Corrected

Comment 12. Please swap Figure 6 and Figure 7.

Response: Corrected

Comment 13. Line 530 - please capitalize "smiles".

Response: Corrected

  1. Lines 576-577 - please correct "invitro and invivo".

Response: Corrected

Reviewer 3 Report

Comments and Suggestions for Authors

The current work "Exploring potential HDAC’s inhibitors for halting cancer pro-gression: An insight of virtual screening, molecular docking coupled with molecular dynamic simulations" is interesting and the contents are informative. The authors investigated This study seeks to discover new inhibitors that selectively inhibit HDAC’s which are linked to deadly disorders like T-cell lymphoma, childhood neuroblastoma and colon cancer. They characterized the structures using in silico methods. I can recommend this work for publishing but I still advice the authors to revise their work.

1- How you selected the lead compound; you may kindly explain it more.

2- About the toxicity issues of the derivatives; you may kindly explain it more.

3- Drug-ability of compounds is an important issue; you may kindly explain it more.

4- For the case of exploring inhibitors, you may include the following references:

https://doi.org/10.1016/j.jpha.2023.04.015

https://doi.org/10.1080/14756366.2021.2016734

https://doi.org/10.1152/ajpgi.00061.2018

https://doi.org/10.3892/ijmm.2017.2989

6- For the case of Docking simulations of derivatives, you may include the following references:

https://doi.org/10.1002/slct.202303989

Author Response

Thank you very much for your thoughtful and encouraging comments on our manuscript titled "Exploring potential HDAC inhibitors for halting cancer progression: An insight of virtual screening, molecular docking coupled with molecular dynamic simulations." We greatly appreciate your positive assessment of our work and your recommendation for publication. We have carefully reviewed your suggestions and recommendations for revisions, and I'm pleased to inform you that we have incorporated all of your comments and references into the revised version of the manuscript. We believe that these revisions have strengthened the overall quality and coherence of the paper. Your insights have undoubtedly contributed to the refinement of our research, and we are grateful for your time and expertise.

Please find attached the revised manuscript for your review. Thank you

REVIEWER 3

Comment 1:  How you selected the lead compound; you may kindly explain it more.

Response: Thank you for the valuable comment. Initially, the drug-like library was docked against the targeted enzymes. The lead compounds were then selected based on two criteria: the compound exhibiting the minimum binding score and the interaction mechanism in the active site of the targeted HDAC enzymes along with the chelation of Zn2+ ions in the catalytic sites of the proteins. The selection criteria were explained briefly in the manuscript as per your comment.

Comment 2:  About the toxicity issues of the derivatives; you may kindly explain it more.

Response: Thank you for the comment. The toxicity of the compounds is explained briefly and incorporated in the manuscript.

Comment 3:  Drug-ability of compounds is an important issue; you may kindly explain it more.

Response: Thank you for the comment. The drug ability and biological activities of the compounds are explained briefly and incorporated in the manuscript using an additional Pass online server.

Comment 4:  For the case of exploring inhibitors, you may include the following references:

Response: Thank you for the comment. The references you suggested for exploring inhibitors are cited in the manuscript.

Comment 5. For the case of Docking simulations of derivatives, you may include the following references:

Response: Thank you for the valuable comment. The references you suggested for Docking simulations are cited in the manuscript.

Reviewer 4 Report

Comments and Suggestions for Authors

The manuscript entitled “Exploring potential HDAC’s inhibitors for halting cancer progression: An insight of virtual screening, molecular docking coupled with molecular dynamic simulations” utilized the molecular docking method to screen HDAC inhibitors. Although the development of HDAC inhibitors is critical for cancer therapeutics and a lot of simulations have been carried out in the manuscript, no real enzymatic experiment was produced with the best candidates to prove their HDAC inhibitory activity, thus significantly lowing the novelty and the value of the molecular docking results. In addition, the manuscript was presented with massive errors and formatting issues. The following suggestions may help to improve the manuscript.

1.      The introduction part briefly introduces the HDAC families and inhibitors. However, the development of HDAC inhibitors is not very clear according to the context, and it’s suggested to list the most representative HDAC inhibitors in a table or figure with structures or other information.

2.      Please provide the chemical structure for compound LIG1-8 in Table 1.

3.      Massive formatting issues throughout the context should be resolved:

(a)    Some texts are abnormally large in the manuscript.

(b)   A bank between units and values should be noticed.

(c)    The ion should be correctly written as Zn2+.

Author Response

Thank you for your feedback on our manuscript titled "Exploring potential HDAC inhibitors for halting cancer progression: An insight of virtual screening, molecular docking coupled with molecular dynamic simulations." We appreciate your time and effort in providing constructive suggestions for improving our work. We acknowledge your concern regarding the absence of real enzymatic experiments to validate the HDAC inhibitory activity of the identified candidates. As computational bioinformatics analysts, our primary focus is on utilizing computational methods to screen and predict potential inhibitors based on binding scores and interaction patterns. While we recognize the importance of experimental validation, our study aimed to provide insights into the binding mechanisms and stability of the identified compounds through molecular docking and molecular dynamic simulations. Furthermore, we have carefully addressed all the comments provided, including formatting issues and errors in the manuscript. Additionally, we have thoroughly reviewed the text to ensure clarity and coherence, while also rectifying any grammatical mistakes. We believe that the incorporation of your suggestions has enhanced the quality and integrity of our research. We appreciate your guidance and input, which have undoubtedly contributed to the refinement of our manuscript. Please find attached the revised version of the manuscript for your review. We look forward to receiving any further feedback or recommendations you may have.

Thank you once again for your valuable insights and for your consideration of our work. 

Comment 1: The introduction part briefly introduces the HDAC families and inhibitors. However, the development of HDAC inhibitors is not very clear according to the context, and it’s suggested to list the most representative HDAC inhibitors in a table or figure with structures or other information.

Response: Thank you for your insightful comment regarding the clarity of the development of HDAC inhibitors in the introduction. I appreciate the opportunity to address this concern. Our study aims to identify the drug-like inhibitor from the Zinc library. Selection of inhibitors based on two main criteria the binding mechanism in the active site and chelating with Zinc ions followed by the highest binding score. In this study, the selected ligands, LIG 1, LIG 2, and LIG3 exhibited the highest binding score and interaction patterns and were further validated by MD simulation. The current finding proposed that the selected inhibitors LIG1 and LIG2 offer promising prospects as HDACI’s inhibitors for cancer treatment, although further investigation invitro and invivo will be needed to validate their functional activity.

Comment 2:  Please provide the chemical structure for compound LIG1-8 in Table 1

Response: Thank you for your valuable comment the chemical structure of the ligands was incorporated in Table 1.

Comment 3:   Massive formatting issues throughout the context should be resolved:

Response: Thank you for your valuable comment. The formatting issues are solved and modified as per your suggestion in the manuscript.

  • Some texts are abnormally large in the manuscript.

Response: All the texts were modified in the same font.

  • A bank between units and values should be noticed.

Response: Corrected in the manuscript as per your comment.

  • The ion should be correctly written as Zn2+.

Response: Corrected

Round 2

Reviewer 2 Report

Comments and Suggestions for Authors

Authors have successfully corrected all the problems I previously pointed out. The manuscript is ready for publication. Dear authors, I wish you success in your future research!

Author Response

Thank you for the response and consider our manuscript for publication.

Reviewer 3 Report

Comments and Suggestions for Authors

This revised work is suitable for publishing. 

Author Response

(The authors gave the same response as above.)
